# Technology Features of Diamond Coating Deposition on a Carbide Tool

Evgeny Ashkinazi [1], Sergey Fedorov [2,*], Alexander Khomich [3], Vladimir Rogalin [4], Andrey Bolshakov [1], Dmitry Sovyk [1], Sergey Grigoriev [2] and Vitaly Konov [1]

[1] Prokhorov General Physics Institute of the Russian Academy of Sciences, 38 Vavilov st., 119991 Moscow, Russia
[2] Moscow State University of Technology "STANKIN", 1 Vadkovsky per., 127055 Moscow , Russia
[3] Fryazinsky Branch of the V.A. Kotelnikov Institute of Radio Engineering and Electronics of the Russian Academy of Sciences, Moscow region, 1 Vvedensky Square, 141190 Fryazino, Russia
[4] Institute of Electric Phisics and Electric Power Engineering of the Russian Academy of Sciences, Dvortsovaya quay, 191186 St. Petersburg, Russia
* Correspondence: sv.fedorov@icloud.com; Tel.: +7-(916)-290-2607

**Abstract:** The production of carbide tools with polycrystalline diamond coatings, which are used for processing modern carbon composite materials, includes a number of technological techniques that ensure reliable adhesion of the coating to the substrate. This review examines these features of substrate-surface pretreatment to improve adhesion, which includes chemical etching, mechanical hardening, modification by ion beams, plasma treatment and application of buffer layers between the substrate and the coating. This review also discusses the advantages and disadvantages of the most common methods for obtaining polycrystalline diamond coatings using hot filament and deposition of coatings from microwave plasma.

**Keywords:** carbon fiber reinforced polymer materials (CFRP); polycrystalline diamond coatings; tungsten carbide; adhesion; buffer layer; diamond tools; hardness; wear resistance; CVD

## 1. Introduction

With the advent of new materials, the question of their machinability inevitably arises. Carbon Fiber Reinforced Polymer materials (CFRP) are characterized by highly specific stiffness and strength, low weight, high corrosion resistance, and a low coefficient of thermal expansion (CTE). They are widely used for various engineering applications, especially in the aviation and aerospace industries [1–4].

Traditionally, the machinability of hard-to-process materials increases due to wear-resistant coatings [5]. In the case of CFRP, the diamond-coated tool has proven itself in the best way. Progress in the creation of diamond tools with increased wear resistance is associated with the need to overcome the technological barrier for the existing tool, initiated by the rapid development of new materials with unique properties designed to work in critical operating conditions. Overcoming the frame's super-strength combination with the bundle's superelasticity—subject to one of the main requirements—the rejection of using coolants during processing is possible only with record-low friction coefficients and maximum thermal conductivity characteristic of a diamond coating.

Diamond has unique properties (high hardness and thermal conductivity, excellent abrasion resistance, low coefficient of friction [6,7]) and is widely used for processing materials. The development of methods for applying durable polycrystalline diamond coatings (DC) directly to the working surface of the base materials makes it possible to manufacture tools using the best properties of diamond. The most common method of applying DC to hard alloys based on tungsten carbide (WC-Co) is deposition from the gas phase by chemical vapor deposition (CVD) [8]. Nevertheless, obtaining a solid layer of diamond coating on the surface of a hard alloy remains a difficult technological task.

To date, such a tool is the most promising substitute for a cutting tool based on poly-crystals of diamond (PCD) obtained by diamond powder sintering. The main difference is the method of diamond synthesis. CVD diamond coating with a thickness of 10–20 μm has a denser polycrystalline structure with a tensile strength in the range of 7000–12,000 kg/mm$^2$ and a friction coefficient of 0.05–0.15. Often, the hardness of the coating is 15% higher than PCD. The advantages of diamond coatings also include the possibility of their synthesis on an instrument of complex shape. PCD tools, due to technological limitations, are made in the form of plates only.

The typical cutting speeds for a diamond–coated tool are in the range of 200–2000 m/min, while the operating speed range of the PCD tool is 780–900 m/min.

Figure 1 shows a schematic diagram of obtaining DC on WC-Co alloy [9]. However, numerous experiments have shown that it is practically impossible to get a high-quality diamond coating with good adhesion by direct deposition. Cobalt catalytically interacts with carbon atoms at a diamond deposition temperature of 600–900 °C and stimulates the formation of a softening graphite intermediate layer at the coating–substrate interface [7,10], which noticeably worsens the adhesion of the DC to the substrate.

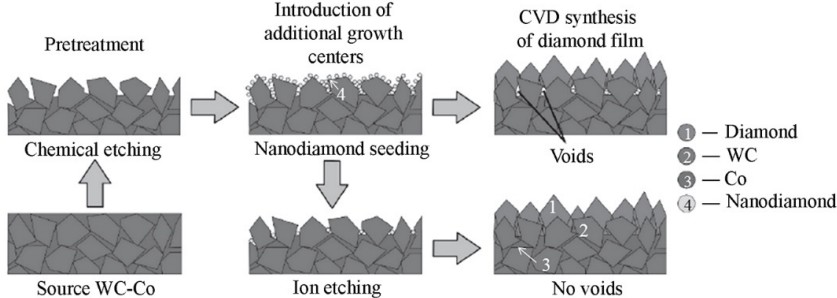

**Figure 1.** Schematic diagram of the production of diamond films on WC-Co alloy [6].

As a rule, the service life of the DC is determined by the peeling of the coating, not by its wear [11]. Several types of substrate surface pretreatment have been developed to solve the adhesion problem, including the preliminary chemical, mechanical, laser, or plasma treatments, as well as the creation of intermediate layers between the DC and the substrate [12–15]. After chemical etching, a middle buffer layer must apply to the substrate to inhibit the diffusion of Co to the inner surface and dampen the difference in the substrate CTE and the diamond, causing the appearance of significant mechanical stresses in the coating layer that can lead to peeling.

Table 1 shows the CTE values of some materials suitable for creating buffer layers. Unfortunately, there are practically no materials close to diamond in terms of CTE, except invar (an alloy of nickel and iron), which is not suitable for this task since Fe, similar to Co, chemically interacts actively with carbon. In the works [16–20], the choice of tungsten as a buffer layer material is justified by its smaller CTE compared to other metals.

**Table 1.** Coefficient of thermal expansion values of materials suitable for making the buffer.

| Substrate | CTE $10^{-6}$ sm$^{-1}$ (0 °C) |
|-----------|-------------------------------|
| WC | 3.8 |
| WC-Co | 4.6–5 |
| W | 4.3 |
| Co | 12 |
| Ta | 6.5 |
| Cr | 5.8 |
| Diamond | 1.2 |
| Si | 5.1 |
| Graphite | 1.5 |
| TaC | 4.6 |

First of all, the surface of the tungsten-containing hard alloy is maximally cleaned of cobalt by chemical etching. Unfortunately, at the same time, the strength properties of the cobalt-depleted layer inevitably deteriorate, and it is desirable to strengthen it, for example, mechanically. Then a buffer layer is applied to the hard alloy, and its surface is seeded with nano-sized diamond particles, which are necessary as a seed for growing DC.

This paper presents an overview of the prominent publications on the technological processes of manufacturing tools based on tungsten carbide with diamond coating and preparing a carbide substrate for its deposition.

## 2. Chemical Methods of Etching Cobalt on the Surface of WC-Co Substrate

The chemical methods of etching the surface of WC-Co substrate are used to remove cobalt from its near-surface layer to increase the density of diamond nucleation and improve the adhesion of DC to WC-Co alloy [21]. The etching procedure is as follows: first, the surface layer is treated with a Murakami reagent ($K_3[Fe(CN)_6]$:KOH:$H_2O$ in a ratio of 1:1:10), and then in a solution of peroxide mono sulfuric acid $H_2SO_5$ (Caro's acid) [22–29].

The Murakami reagent selectively etches tungsten carbide grains and removes cobalt from the surface layer while creating the necessary surface roughness of the hard alloy. Caro acid eroded cobalt to soluble $Co^{2+}$ compounds, reducing the concentration of Co on the surface [30] and oxidizing a layer of super stoichiometric carbon, interfering with adhesion in amorphous form ($sp_2$) to carbon dioxide gas. An increase in surface roughness also increases the effective area, which contributes to an increase in DC adhesion [15]. At the same time, careful monitoring of the thickness of the etched zone is necessary since the residual pores formed as a result of etching reduce the adequate thickness of the layer, providing adhesion [31] and reduces the viscosity of the near-surface area, which causes embrittlement of a hard alloy [30,32].

Selective etching Co with acids involves a wide range of different pretreatment methods. The Murakami and Caro's reagent treatment time depends on the WC grains size and the Co% concentration [15,33]. Moreover, additional factors must be considered to obtain good adhesion of DC with WC-Co [26,34].

The adhesion of DC to the carbide base weakens due to the presence of a certain number of micropores at the grain boundaries, the graphite phase, and high residual stresses at the DC–WC interface [35]. The authors [26,35] recommend using WC-Co with a low Co content and paying attention to the sufficient roughness of the substrate because Co catalyzes the synthesis of graphitized $sp_2$-bound carbon rather than the formation of diamond $sp_3$-carbon and leads to the growth of non-diamond (graphite) phases acting as softening layers [34–36], which reduces the deposition rate of diamond [35].

There are three main ways to neutralize the effect of cobalt on the adhesion of DC:

- The coating of an intermediate layer can prevent the diffusion of Co to the surface of the substrate;
- The creation of stable intermetallic cobalt compounds. This method is implemented using gas or liquid phase reactions;
- The removal of cobalt from the surface of a hard alloy [37,38] by laser ablation, selective dissolution of cobalt with aggressive chemicals, and by combined heat treatment (heating WC-Co at 1000 °C in an atmosphere of 0.25% $CH_4/H_2$) with subsequent selective removal of cobalt in Caro acid.

Another way to remove Co from the WC-Co surface is to treat it with microwave $CO_2$–$O_2$ plasma at 900 °C. As a result of plasma action, $CoWO_4$ and $WO_3$ compounds are formed. These phases are removed in an alkaline solution [25,39].

The advanced chemical treatment used before the DC coating to WC-Co is a two-stage method based on the etching of WC-Co substrates in the Murakami reagent followed by the dissolution of Co by selective etching in Caro acid [27,39–41].

In ref [42], the results of various chemical treatment regimes were compared to identify the most effective process of Co etching. Three modes with other concentrations of reagents and different etching times were selected for Murakami etching. The concentration

of $K_3[Fe(CN)_6]$ varied in the range of 10, 20, and 30 wt.%. The concentration of KOH varied in the range of 10, 20, and 30 wt.%, and the etching times were 10, 15, and 20 min. A total of 11 experiments were conducted. For the second stage using HCl, three parameters were also changed: HCl concentration (40, 53, 65%), $H_2O_2$ concentration (10, 21, 32%), and etching time (10, 15, 20 min). As a result, it has been shown that—to develop more roughness—it is necessary to increase the etching time to 20 min in a Murakami mixture of 30 wt.% $K_3[Fe(CN)_6]$ and 10 wt.% KOH. The most effective for complete (0%) removal of cobalt from the surface was etching for 1 h in a modified mixture of Caro (65 vol.% $HNO_3$ + 32 vol.% $H_2O_2$) in an ultrasonic bath.

In ref [43], the effect of the substrate surface pretreatment on the cutting efficiency of the processed material during the micro treatment of a carbon CFRP with a WC-Co (10%) tool coated with micro crystalline diamond was experimentally investigated. Etching in the Murakami solution for 120 min was the most effective way to remove Co and improve the quality of microcrystalline DC.

In ref [44], the effect of etching duration on the adhesion of DC on WC-Co drills (6 wt.%) was studied. The drills were first subjected to ultrasonic treatment in a mixture of 10 mL of 98% $H_2SO_4$ + 30 mL of 38% $H_2O_2$ for 10, 20, 30, 60, and 90 s. The process was carried out at a temperature of 120 °C to improve the etching efficiency. The removal of Co was accompanied by the appearance of a light pink liquid, characteristic of Co salts, so all samples were thoroughly washed with distilled water in an ultrasonic bath until a clear liquid was obtained. The drills were treated in an ultrasound with a solution of ultrafine detonation diamond nanoparticles of 8–10 nm in size to increase the density of diamond nucleation. Raman scattering spectra showed that a high-quality microcrystalline DC was successfully grown after etching for 30 s. An increase in the duration of preliminary processing led to the appearance of a porous structure, island films of DC, and deterioration of the operating parameters of the drills.

In ref [45], the process of preliminary chemical treatment of WC-Co alloy before DC deposition was optimized. The size of WC grains, which significantly affects the thickness of the etched Co layer, was determined depending on the time of the process in two ways: by determining the mass of dissolved Co and by the depth of profiling of the etched surface. Although the removal of Co is beneficial, it causes a negative effect of WC-Co surface layer softening, during which a porous tungsten carbide skeleton forms [15].

The authors of the work [46] used the 6% Co alloy, initially purified for 10 min in an ultrasonic bath with ethanol. The treatment of the hard alloy is carried out with Murakami solution in an ultrasonic bath, followed by washing with deionized water. Cobalt was removed from the surface of the hard alloy with Caro's acid for 7, 11, or 15 min. Next, nanodiamond nuclei were seeded on the surface of the hard alloy in a diamond suspension in an ultrasonic bath. The authors believe that the etching of a hard alloy in a Caro's solution has little effect on its roughness. The best results demonstrated the treatment mode in Murakami solution for 7 min + Caro's acid for 11 min for the WC-6 mass % Co. It is necessary to optimize the acid etching time to improve the quality of the DC.

Thus, in the works [21–46], the modes of preliminary chemical treatment of a hard alloy were established, ensuring the best adhesion of the tool with the applied DC.

## 3. Chemical Treatment of WC-Co Substrates with a High Cobalt Content

Most studies of the adhesion of DC to hard alloys were performed on substrates with a low concentration of Co (up to 6 wt.%). However, for hard-to-process materials, a tool with a cobalt concentration of more than 6 wt.% is required. In ref [47], the pretreatment processes of WC-9% Co substrates were investigated. This process is based on two-stage chemical etching followed by seeding with diamond nanoparticles and studying the methods of nucleation and growth of DC. For such substrates, the authors proposed to reduce the etching time of a hard alloy in a Murakami solution to 10 min with simultaneous exposure to ultrasound, which is sufficient for the manifestation of WC granularity and time of Co

removed at the second stage of etching in Caro solution. The optimal time of the second etching stage is 5 min.

In ref [48], the etching processes of a tungsten carbide-based hard alloy with a Co content of 6, 10, and 12 wt.% were experimentally studied. The transformations of Co during treatment in Murakami solution and DC deposition by the hot thread method were investigated, including nucleation, growth, and heating of processed materials without a carbon source. A two-stage pretreatment method was used: ultrasonic etching in a Murakami solution with different dilution levels and duration of stay in an ultrasonic bath and etching of the sample using Caro's solution with varying levels of dilution. An additional processing stage is proposed to increase the concentration of nucleation sites. This stage consisted of wiping the sample with a paste of diamond chips (3 microns) in glycerin to remove the loose surface layer.

As a result of the research, a technological scheme was proposed that includes seven main stages and describes the behavior of Co in the manufacture of a tool with DC, taking into account the stage of pretreatment, nucleation, and growth of the diamond coating. The mechanisms of Co diffusion in the hard alloy surface layer when heated in a hydrogen atmosphere and the evolution of Co compounds in an untreated solid-alloy tool with the same parameters of nucleation and heating processes considered. Relatively high substrate temperatures (800 and 950 °C) will stimulate (due to diffusion from the volume to the surface) an increase in the cobalt concentration on pretreated surfaces and negatively affect the rate of nucleation and the quality of DC. For untreated samples at a very high temperature (950 °C), the Co content on the substrate surface decreases, probably due to the sublimation of Co. During deposition, a dynamic balance is formed between the diffusion of Co from the volume to the surface and the subsequent substitution of Co, while the contribution of diffusion prevails. However, the diffusion of Co from the interface to the DC gradually decreases and even disappears because of the deposited diamond crystals' barrier effect with an increase in the growth duration. Thus, Co diffusion is a factor that cannot prevent the formation of embryos, but the growth parameters and DC quality near the film–substrate interface are comparatively worse. The depth of removal of Co from the surface layer for WC-Co alloys with a high Co content should be at least 8–9 mkm. It is also necessary to use not only etching but also other operations of preliminary processing of WC-Co substrates, as was proposed, for example, in ref [48].

## 4. Physical-Mechanical Methods of W-Co Alloys Surface Pretreatment

Before the application of DC, several WC-Co treatment methods were used to increase adhesion, including chemical processes, mechanical treatment, plasma methods, and laser ablation. Thus, in ref [49], the high efficiency of impact hydrotreatment for improving the adhesion of DC on the WC-Co surface is shown. Mechanical treatment methods often remove Co from the WC-Co surface [50]. In ref [51], a complex pretreatment method was tested, including sandblasting, two-stage pretreatment, shot blasting with solid particles ($SiO_2$ or $Al_2O_3$) and ultrasonic cavitation, and nanocrystalline DC deposited on the surface of a hard alloy treated in this way. Before DC deposition, a Cr/CrN/Cr buffer layer is applied to the hard alloy [52]. Subsequent mechanical treatment with diamond powders increased the roughness of the hard alloy. It increased the effective surface area of the coating deposition, which also increased the nucleation density necessary to accelerate the deposition of CVD diamond and increased the adhesion of DC. However, the thick buffer layer reduces the efficiency of the instruments with DC cutting edges.

In refs [25,50,53,54], the effect of laser and plasma treatment and the adhesion of the coating to WC-Co were investigated. Laser treatment of the surface provided adhesive strength comparable to the preliminary microblasting. Pulsed laser irradiation creates an optimal surface texture for good adhesion of the DC [55].

Figure 2 shows images of diamond coatings on the surface of a hard alloy.

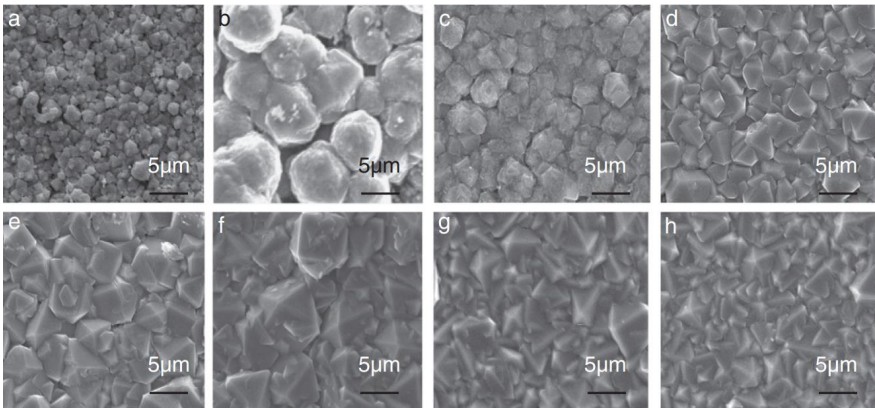

**Figure 2.** SEM images of diamond coating deposited on WC-Co substrates pretreated by: (**a**) abrading; (**b**) Cr interlayer deposition; (**c**) Nb interlayer deposition; (**d**) Ta interlayer deposition; (**e**) chemical etching; (**f**) chemical etching and Cr interlayer deposition; (**g**) chemical etching and Nb interlayer deposition; and (**h**) chemical etching and Ta interlayer deposition [28].

Ref [56] presents the results of a preliminary combination of a WC-Co treatment with a high Co content, which combined chemical etching and sandblasting, and created a carbide-forming metal layer (Figure 3). The adhesion of DC to a hard alloy was higher than when using only chemical etching or applying a sublayer. The carbide-forming metal forms related carbides during DC growth. Despite the high temperature required to accelerate the diamond nucleation process, carbides prevent the diffusion of Co and suppress the forming graphite phase. The layers of pure Cr, Nb, and Ta are usually used in this case.

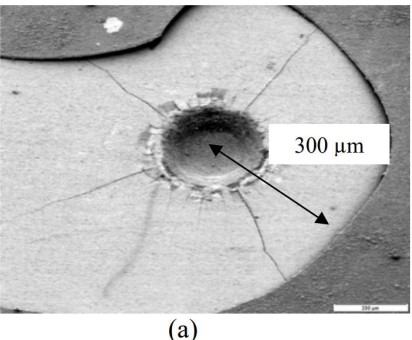
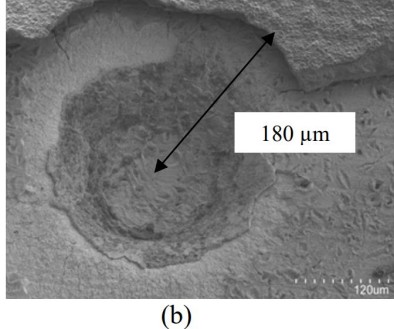

(a)                                    (b)

**Figure 3.** Effect of sandblasting with diamond powder on Cr-coated WC-Co substrate: (**a**) delamination area without sandblasting and (**b**) the elamination area with sandblasting [56].

In ref [27], it has been shown that chemical etching and the CrN/Cr buffer layer contribute to an increase in the adhesion of DC. It has also been found that after the chemical removal of Co from the surface of a hard alloy, the deposited chromium not only prevents the diffusion of Co but also improves the porous structure resulting from chemical etching, which increases the adhesion of DC [28]. This result is of considerable interest, as it confirms the unique adhesive properties of chromium, which is widely used as a sublayer when applying a variety of coatings.

In ref [57], the chemical etching pretreatment, laser etching, and acid treatment of WC-Co, followed by laser etching, were investigated. Laser irradiation leads to a periodic relief of the irradiated surface, which increases its roughness and reduces mechanical stress between the substrate and the deposited DC. The properties of the DC and the magnitude of the residual compression stresses in the coatings were studied by RAMAN spectroscopy. Chemical etching was used to remove Co from the surface, but the surface roughness created by etching did not sufficiently reduce mechanical stresses in the DC. In constant, laser etching reduced the number of mechanical stresses, but the concentration of Co

remained high. The combination of chemical etching and laser treatment methods ensured the production of a homogeneous high-quality DC, high adhesion of the coating to the substrate, and a low residual stress level.

Plasma modifies only a few atomic layers of the substrate without changing the bulk properties of the hard alloy [52,58,59]. In ref [60], pooled plasma and chemical etching were used to prepare WC-Co before the deposition of (TiAl)N layers by vacuum spraying. Plasma etching increases the surface roughness and slightly reduces the Co content. It concluded that the processing modes studied in this work, including plasma etching, are suitable for applying a sublayer (TiAl)N to a WC-Co tool, increasing the cutting time by up to 59%.

In ref [61], sandblasting + acid etching was used for the pretreatment of WC-Co before the deposition of DC. The surface roughness varied depending on the parameters of sandblasting. The porous Co-less layer after sandblasting and acid etching was thinner than after the two-stage chemical treatment.

The effect of surface sandblasting on nucleation during chemical gas-phase deposition of the coating has also been studied. Mechanical tests have shown that sandblasting increases the adhesion of the coating compared to two-stage chemical treatment. X-ray diffraction analysis and RAMAN spectroscopy have recorded a decrease in residual stresses in the DC; tests on a cutting machine have shown that tools with DC that have been sandblasted and etched in acid have a smaller wear area of the flank and coating. Thus, in ref [61], it was concluded that sandblasting is a stable, safe, and inexpensive technology of the pretreatment for the tool preceding the CVD process.

## 5. Suppression of Co Diffusion and Increasing of DC Adhesion by Buffer Layers Formation

The etching of the surface of a hard alloy is not consistent with the tendency to increase the Co content in WC-Co to improve the properties of the carbide tool. The use of etching alone is ineffective if the Co content in the hard alloy exceeds 6 wt.%. In addition, at a high deposition temperature, cobalt from the volume can diffuse to the etched surface [62]. Therefore, to suppress the carbon-cobalt reaction during the deposition of the diamond coating by the CVD method, buffer layers are used between the carbide substrate and the DC [21,63–69]. A properly selected intermediate (buffer) layer serves as a diffusion barrier for Co, which can improve the adhesion of DC to WC-10% Co substrates.

The buffer layer structure affected the adhesion of DC to a hard alloy [70]. In ref [71], the buffer layers TiN, TiC, (TiSi)N, $Si_3N_4$, a-SiC, and a-$SiC_xN_y$ were investigated, and it was found that DC has insufficient adhesion to TiN, and a pulsed arc deposited an a-C (laser-arc). The DC has the best adhesion to the SiC and $Si_3N_4$ layers due to the best correspondence of the CTE gradients. In ref [21], a comparative study of CrN, ZrN, NbN, and TaN (with a thickness of 1 mkm as barrier layers for Co diffusion during DC deposition at 880 °C on WC-12 wt.% substrates) shows that NbN and TaN interact with carbon during DC deposition, forming carbides, and are the most effective barriers. A small diffusion depth of Co and weak carburization of the buffer layer were observed when using the ZrN layer, and the CrN layer was wholly transformed due to carburization.

Co-diffusion blocker's amorphous layers demonstrate high values of yield, tensile strength, and corrosion resistance [72,73]. In ref [71], amorphous layers a-SiC, a-$Si_3N_4$, and a-$SiC_xN_y$ were synthesized by CVD, on which diamond coatings were deposited. It established that amorphous buffer layers of SiC improve adhesion, friction properties, and the cutting efficiency of DC by WC-6 wt.% Co [74,75].

Improved adhesion to the substrate is also achieved after pretreatment, which leads to the formation of stable Co compounds, such as borides or silicides, which reduce the pressure of Co vapors and mitigate the effects of the volatile Co phase [76]. Various metals and ceramics were also used as buffer layers: tungsten [16–20,67,77], aluminum [78,79], chromium [80], NbC [81], $Si_3N_4$ [82], TiN, Ti(CN), and CrN [21,81,83]. In ref [69], good DC adhesion with a thickness of 8 and 20 microns was demonstrated using a CrN buffer layer.

The buffer layers blocking Co diffusion methods are not limited only by physical (PVD) or chemical vapor deposition (CVD) technologies [53]. It is known that some intermediate layers hinder the nucleation of the diamond and inhibit the growth of DC. In this case, the additional WC-Co treatment increases the density of diamond nuclei [84] and complicates the entire technological process.

In refs [85,86], CrN was proposed as the optimal buffer layer material for DC on WC-10% Co substrates, which contradicts the results of ref [21]. The use of CrN effectively prevents the diffusion of Co from the substrate, preventing the catalytic activity of Co during the deposition of DC, which ensures good chemical and mechanical adhesion to DC due to the formation of a thin layer of CrC [85–87]. It is shown in ref [88] that the adhesion of DC and WC-10% Co with a Cr-N interlayer sharply depends on the diamond layer deposition temperature. In addition, the formation of a chemical bond at the interface and residual stresses in the films have opposite temperature dependences, which makes it possible to optimize the deposition temperature of the diamond-based coating for better adhesion. Though less strong adhesion was observed at the diamond/Cr-N interface.

The effect of a diamond coating deposition temperature on adhesion to WC-10% Co substrates with a Cr-N buffer layer was studied in ref [77]. The DC was deposited at various temperatures (550, 650, and 750 °C) in a CVD reactor with a hot thread. It was found that the carbon diffusion through the diamond/CrN film significantly depends on the deposition temperature and leads to partial carburization. Nitrogen in CrN is replaced by carbon, and a CrC layer is formed. Its formation is influenced by both the rate of growth (coalescence) of alkali grains and the rate of diffusion of carbon into the interlayer, which have different temperature dependences. The rate of DC growth linearly depends on the deposition temperature [88], and the rate of carbon diffusion has an exponential temperature dependence $D = D_0 exp(E_a/kT)$. As the substrate temperature increases, the DC growth rate increases, which reduces the duration of the carburization process by replacing Cr-N with Cr-C, while the rate of carbon diffusion into the Cr-N layer also increases exponentially with temperature. This leads to the fact that carbon diffuses to a greater depth, forming a thicker Cr-C layer before the diamond grains form a continuous film. Over time, the concentration of active carbon trends to zero due to the coalescence of diamond grains and the formation of a solid diamond film, which limits the supply of reagents for further carburization. In addition, the density of the Cr-C layer formed at the interface ($6.68 g/cm^3$) is higher than that of Cr-N ($5.9 g/cm^3$), which also leads to a decrease in the diffusion rate of carbon and nitrogen. The formation of a Cr-C layer provides a strong chemical bond between the DC and the Cr-N layer, which contributes to its adhesive strength. At the same time, an increase in the Cr-C layer thickness is undesirable since the difference between the CTE of the diamond and $Cr_3C_2$ is higher than in the diamond–Cr-N pair. According to the results of Rockwell hardness measurement, the best adhesion was obtained for films applied at 650 °C. At deposition temperatures of 550 °C and above 750 °C, the adhesion of the DC was worse. In ref. [88], this is explained by the fact that, at a temperature of 550 °C, too thin of a layer of Cr-C grows. In contrast, at 750 °C, adhesion deteriorates due to residual thermal stresses. A Cr-C layer of optimal thickness is formed at a temperature of 650 °C. It guarantees the absence of cracks on the DC surface or at the surface–buffer layer interface. This does not extend to the Cr-C/Cr-N interface, which prevents the peeling of the coating.

In ref [89], the effect of magnetron sputtering CrN and TiN layers on the adhesion of DC film to WC-Co substrate was investigated. Opposite data about the effectiveness of buffer CrN layers was obtained. DC synthesis was carried out in a CVD reactor with a glow discharge. DC adhesion was determined by Rockwell hardness measurement results. It was found that TiN does not react with DC, but CrN (according to small-angle X-ray diffraction) almost entirely turns into chromium carbide $Cr_3C_2$. However, adhesion tests had shown that the efficiency of the buffer layers studied is significantly lower than when the substrate was pretreated with Murakami and Caro solutions. It concluded that using only the TiN layer to increase the adhesion of the DC is impractical due to the lack of

chemical bonds between the DC and TiN layers. The use of the CrN barrier layer also proved ineffective due to the conversion of chromium nitride into chromium carbide and the embrittlement of DC.

In ref [90], comparative studies were carried out on the adhesion of (111)-textured DC deposited by the hot thread method on buffer layers TiN, CrN, TiC, and SiN deposited on mirror-polished substrates from WC-Co. The adhesion of the coatings was determined by the Vickers microhardness measurement method. The best results were obtained by a sample with a buffer layer of CrN 800 nm thick, pretreated by chemical etching for 10 min. It is shown in ref. [91] that $Cr_2O_3$ is also an effective blocker of Co diffusion, preventing the formation of graphite at the DC/WC-Co boundary. $Cr_2O_3$ was deposited at 300 °C by spraying Cr targets (99.95%) in oxygen and argon plasma. A thin layer of Cr was deposited on it for 5 min to increase adhesion after the deposition of $Cr_2O_3$. This Cr layer allowed a carbide layer to form at the border with DC. At the same time, the need for careful control of the methane dosage in the gas mixture was noted to ensure the optimal concentration of carbon on the surface during the diamond particle nucleation, which is relevant when carbide-forming materials are used as an intermediate layer.

In ref [92], the properties of DC deposited on WC-Co with or without TiN, CrN, TiC, and SiN buffer layers were analyzed. It could be expected that the most significant mechanical stresses in DC would occur when using CrN, TiN, TiC, and, especially, CrN buffer layers, whose KTE differs as much as possible from diamond CTE due to the difference in the CTE of the WC-Co and diamond buffer layers. The maximum residual stresses were observed in the DC with the SiN buffer layer. Microhardness measurements have shown that the SiN buffer layer, which does not chemically interact with the diamond and the substrate, does not provide sufficient adhesion of the DC with the substrate. Another reason for the negative effect of SiN on adhesion may be its amorphous structure, which does not ensure the alignment of crystal lattices according to CTE, and its high hardness. The interaction of TiN and TiC buffer layers with DC increases the adhesion of the coating to the substrate. Still, the maximum positive effect on the adhesion of DC is provided by buffer layers that chemically interact with the diamond. During etching, pits form on the surface of the substrate, and filling these pits on the grain boundaries with a buffer layer leads to mechanical adhesion of the substrate to the buffer layer. This effect was observed on samples with all the buffer layers studied. In contrast, chemical reactions with diamond occurred only in the CrN buffer layer, which provided better adhesion of the DC with the CrN layer.

In ref [48], an increase in the adhesion of DC to WC-Co provided by magnetron sputtering of homogeneous buffer layers of $TiB_2$ with a thickness of about 0.1–1 μm, effectively preventing the diffusion of Co from the substrate into the coating discussed. At the same time, the density of the embryos and the growth rate of the diamond layer increased. An increase in adhesion was observed at the optimal thickness of the $TiB_2$ layer of 200 nm.

In ref. [81], the deposition of DC at 780 °C on $CrN_x$ and NbC layers with a thickness of 1 μm was studied. The choice of these layers was due to their high hardness and proximity of CTE to its value of diamond and WC-Co [93,94]. The minimum level of mechanical stresses in the coatings during its cooling at the stage of the growth process's completion was provided. Studies of the barrier properties of the $CrN_x$ buffer layer have shown that, during heat treatment in a hydrogen atmosphere, a significant amount of Co diffuses through this layer, having a porous structure. Thus, using a $CrN_x$ layer to obtain a DC with good adhesion requires the application of an additional barrier layer that prevents the diffusion of Co.

The properties of the NbC buffer layer significantly differ from the $CrN_x$ layer. The NbC layer is thermally stable when heated in a hydrogen atmosphere, although traces of Co diffusion through NbC was observed after exposure for 60 h. Sufficient adhesion of DC to NbC was achieved only after increasing the roughness of WC-Co before Co poisoning and precipitation of NbC. The appropriate WC-Co pretreatment is necessary to obtain

sufficient adhesion before the deposition of NbC. A significant improvement in adhesion was observed after the Murakami reagent or sandblasting + Murakami reaction. In ref [81], it concluded that a double buffer layer of $Cr_xC_y$ and $CrN_x$ can be optimal for the successful application of DC. The $Cr_xC_y$ layer is a diffusion barrier for Co, and the $CrN_x$ layer, after transformation into a porous structure, provides good adhesion of the DC.

## 6. Two-Layer and Gradient Buffer Layers

In ref [95], dense and uniform gradient coatings from $Mo/Mo_2C$ layers were synthesized on WC-Co substrates by their two-stage ion-plasma treatment in argon with the addition (at the second stage) of methane. The deposition of coatings was carried out on an ion-plasma installation with two cathodes. The formation of $Mo_2C$ occurs due to the reaction of Mo with C in the atmosphere of $CH_4$. Diamond-coating deposition was carried out in a high-frequency (2.45 GHz) plasma CVD reactor. The morphology, phase composition, adhesion of buffer layers, and their influence on the subsequent deposition of DC were studied in this work. A diffusion layer of $Mo/Mo_2C$ with a thickness of ~4.0 μm was applied to the WC-Co nano substrate, and a DC layer with a thickness of ~2.7 μm was deposited on it by the CVD method. Analysis of the distribution of chemical elements by depth showed that the concentration of Mo gradually decreased, and Co was entirely blocked in the diffusion layer and did not affect the formation of DC. The gradient diffusion layer had good adhesion with both the intermediate layer and the nanocrystalline DC since the parameters of the crystal lattices of Mo and diamond are pretty close. Thus, the results of ref [95] demonstrated that intermediate gradient layers of $Mo/Mo_2C$ could be considered buffer layers for applying DC to WC-Co.

Figure 4 shows the cross-section morphology and the distribution of elements over the intermediate layer $Mo/Mo_2C$.

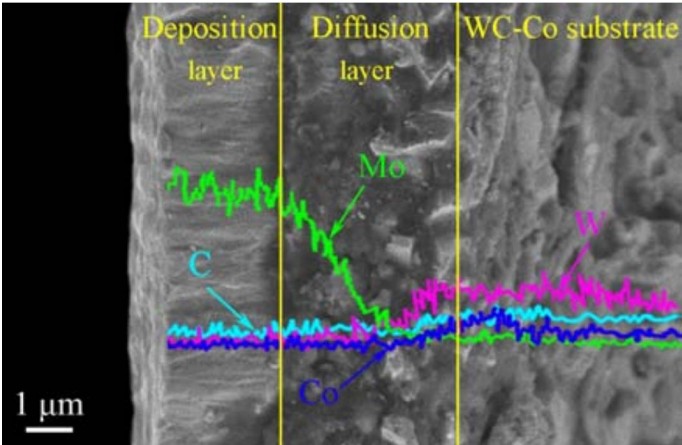

**Figure 4.** Cross-sectional morphology and the corresponding elemental depth distributions in the diamond coating deposited on Mo/Mo2 C sublayer [95].

A two-layer HfC/HfC-SiC diffusion barrier deposited on WC-Co by reactive plasma alloying was also used [96]. The inner nanostructured HfC layer was ~1.0 μm thick, and the outer layer of HfC-SiC had a thickness of ~0.7 μm. After reactive plasma alloying, the microhardness of the surface increased from 1802 to 2857 $HV_{0.1}$. The Co diffusion from WC-Co was effectively suppressed due to the formation of a CoHf and a nanostructured HfC layer. In ref [96], it was concluded that using a diffusion-modified two-layer HfC/HfC-SiC system is one of the most effective ways to increase the adhesion of DC to a WC-Co substrate.

## 7. Buffer Layers Based on Silicon Compounds

The authors [97,98] believe that the cubic carbide of silicon β-SiC can be used as a barrier layer to prevent the diffusion of cobalt. In ref [98], the effect of the SiC buffer layer

on morphology, phase composition, and adhesive strength was investigated. Cobalt reacted with SiC to form the silicides $Co_2Si$ and $CoSi$, which suggests that SiC can suppress the harmful effects of Co without compromising the characteristics of DC. The β-SiC layer reduces residual stresses in diamond coatings, reducing the KTE between WC-Co and the diamond coating [99]. It was shown in ref [100] that hard alloy cutters with DC deposited on the β-SiC buffer layer proved to be more durable than similar cutters hardened with sintered diamond. Using microwave plasma on W, Mo, and WC-Co substrates, diamond/β-SiC nanocomposite structures with increased adhesion were synthesized [101]. The deposition of intermediate layers is provided by a controlled change in the composition of the gas mixture [70,101]. The diamond/SiC interlayers deposited by the CVD method are also used as buffer layers [102]. Composite layers significantly hindered the diffusion of Co from the carbide substrate and allowed the deposition of microcrystalline DC on the buffer layer. The best operational characteristics are achieved with a SiC/diamond phase ratio of about 45% [102].

In ref [103], an improvement in the adhesion of DC to WC-Co substrate was also achieved by using a β-SiC buffer layer. At the same time, it claimed that the β-SiC layer contributed to the origin and growth of DC because, firstly, β-SiC has a diamond-like cubic lattice and, secondly, when activated by atomic hydrogen, $sp_3$ carbon formed from the gas phase due to the substitution of Si atoms in the β-SiC lattice with C atoms. In ref [103], the effect of the deposition temperature and the mode of magnetron sputtering of the β-SiC buffer layer on the processes of DC nucleation and growth was investigated. The magnetron sputtering reduces the temperature forming a buffer layer of silicon carbide. At the same time, it establishes that an increase in the deposition temperature contributes to the elimination of cauliflower-type clusters on the surface of the DC and improves the quality of the β-SiC barrier layer. The β-SiC layer blocks Co diffusion and stimulates the nucleation process during DC deposition. The content of $sp_3$-C bonds in DC increases with the concentration of the β-SiC phase in the buffer layer. DC nucleation and growth on the β-SiC layer can conditionally divide into six stages: initialization, activation, substitution, formation and evolution of embryos, and DC growth. The most critical process determining the appearance of $sp_3$-carbon is the reaction of substitution by C atoms of Si atoms in the β-Si lattice during the activation of $CH_x$ + groups and atomic hydrogen.

In refs [75,104], the influence of amorphous ceramic buffer layers based on silicon on the tribological properties of DC on WC-Co was studied. Amorphous buffer films, a-$SiO_2$ and a-SiC, were synthesized by pyrolysis by CVD on a burning filament of mixtures of tetraethoxysilane, $C_8H_{20}O_4Si$ and dimethyl-diethoxy silane $C_6H_{16}O_2Si$, and then DC was deposited on these layers. For comparison, the DC is worse without an intermediate layer. The grain size and orientation of the crystallites were almost the same in both coatings. The upper layers of the DC on amorphous ceramic layers had less roughness than the coatings applied without an intermediate layer. Films of a two-layer (amorphous ceramic + diamond layer) coating exhibit a lower coefficient of friction than films without a middle layer. As a result, it shows that ceramic buffer layers could act as an amorphous binder to fill the space between diamond grains and brittle WC particles, reducing the surface roughness and improving the DC's tribological characteristics. The a-SiC coating had the lowest surface roughness and the lowest coefficient of friction [75].

## 8. Buffer Layers Based on Tungsten and Tantalum

In refs [16–20], barrier layers of tungsten are applied at the formation stage of diamond nuclei to eliminate the increased diffusional activity of the binder and the clustering of Co. When choosing the method of applying barrier layers of tungsten, it was taken into account that as the layer thickness increases, the internal stresses leading to embrittlement of the barrier layer accumulate. The ion-plasma coating deposition method, when the substrate temperature reaches ~700 °C, is replaced by the cold magnetron deposition when the substrate temperature does not exceed 50–80 °C to prevent cracking of the barrier layer. In this case, the tungsten coating, sprayed at a speed of 1 nm/s, repeats the profile of the

complex substrate relief, and no temperature stresses occur during cooling. At the same time, the adhesion strength is provided by a natural 3D relief formed by a combination of WC grain boundaries. The technology of applying a tungsten film consists of stages using a thin (10–30 nm) tungsten sublayer to a substrate activated by an ion beam at operating pressures of a magnetron discharge in argon (less than 0.25 Pa), with subsequent formation of a tungsten film at a given thickness at pressures of 0.5–0.7 Pa.

The features of the wear mechanism of nanoscale and microcrystalline DC deposited on tungsten carbide substrates were studied by RAMAN spectroscopy [20].

Deposition conditions have a significant influence on the grain size of DC. Figure 5 shows the photographs of the two-layer coating microstructure of micro and nanodiamonds deposited on a tungsten buffer layer at different methane concentrations in a gas mixture.

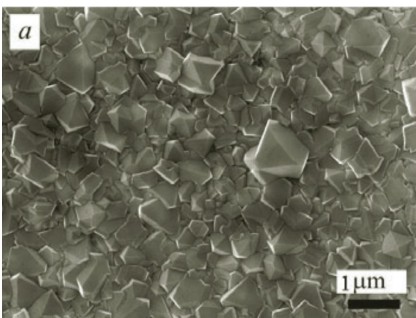 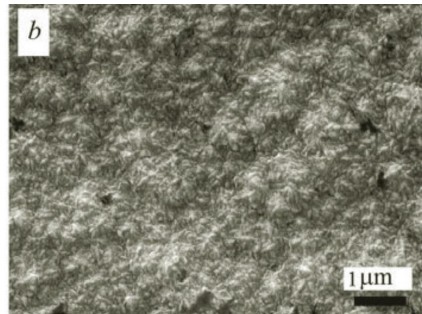

**Figure 5.** SEM images of diamond coatings (DC) deposited at a concentration of methane in the gas mixture: (**a**) 4% (microcrystalline DC) and (**b**) 15% (nanocrystalline DC) [20].

Figure 6 shows the RAMAN spectra of single-layer nanocrystalline, and microcrystalline DC measured in the entire region (spectra 1 and 3) and in the area of maximum wear (spectra 2 and 4) after tribological tests with a counter body ball of $Si_3N_4$ [16–20]. Spectrum 5 in Figure 6 was measured on a natural diamond crystal implanted with nickel ions [105].

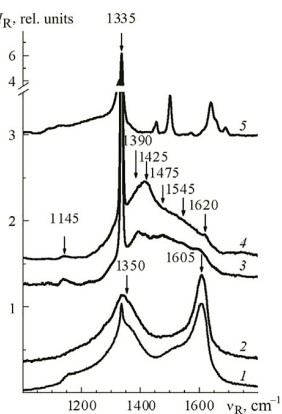

**Figure 6.** Raman spectra of single-layer nano- and microcrystalline diamond coatings were measured in the entire region (1, 3) and the region of maximum wear (2, 4) after tribological tests with a Si3 N4 counter-ball. Spectrum 5 is measured in a natural diamond crystal implanted with nickel ions [105]. The spectra are shifted vertically relative to each other for clarity [20].

In the RAMAN spectra of microcrystalline diamond (Figure 6, spectrum 3), the diamond peak with a maximum at $\Delta \nu R = 1335$ cm$^{-1}$ is dominant. In addition to the low-intensity bands, with maxima at 1155, 1350, and 1475 cm$^{-1}$, it is observed in the spectrum of nanocrystalline DC that relatively narrow bands with maxima around 1390, 1425 and 1620 cm$^{-1}$ appear in the microcrystalline coating (spectra 3 and 4). The relative intensity of these bands varies along the DC surface. It was probably determined by the morphological features of the layer consisting of diamond crystallites with a size of ~0.5–5 μm. A band of about 1620 cm$^{-1}$ in the

spectra may indicate the inclusions of microcrystalline graphite. After tribological tests, the width of the diamond peak increases, and the band 1145 $cm^{-1}$ weakens slightly and shifts to the region of lower frequencies. The band with a maximum of ~1425 $cm^{-1}$ (spectrum 4) is caused by defects in the diamond lattice in the twinning regions.

In refs [21,106], the layers of Ta and its compounds with Cr and Nb were studied as a buffer coating for DC. The nucleation density of diamond particles on the Ta sublayer was the highest, apparently due to the carbide formation. The intermediate layer of Ta effectively suppressed the diffusion of Co. However, the adhesion of Ta to WC-Co was poor due to thermal stresses occurring at the interface with the substrate.

Nevertheless, it is believed that the chemical compound of tantalum with carbon can be used as a buffer layer due to its high hardness and good resistance to thermal shock [107]. It is confirmed in ref [108] that the intermediate layers of $Ta_xC$ and mixtures of $Ta_2C$ and TaC are used. In ref [109], the effect on adhesion of the deposition temperature of DC in microwave plasma (temperature up to 950 °C) on buffer nanocrystalline layers of $Ta_xC$ deposited on WC-Co was studied. It shows that the obtained DC has good adhesion, high hardness, and maximum wear resistance when the coating temperature of tantalum carbide is 800 °C.

A more complex variant of the Ta-based buffer layer with a second layer of ZrN and Mo was proposed in ref [110]. The best results were obtained for a system consisting of 9 periods of a two-layer TaN/ZrN method with a thickness of each layer of 30 nm, on which a two-layer system of TaN (0.5 μm) and Mo (0.5 μm) was applied. When applied, each layer of deposited TaN with a thickness of 100 nm was subjected to ion etching using a reactive mixture of Ar-6 vol.% $N_2$ at a total pressure of 0.02 Pa for 1 h. The sputtering–etching processes repeated until a layer of TaN with a thickness of ~0.5 μm was obtained. The upper layer of Mo (~0.5 μm) was then deposited, providing the highest nucleation of DC. Thus, it shows that the DC's high wear resistance is ensured by preventing the propagation of cracks in a high-hardness, multi-layer nanometer buffer coating.

Some works for tungsten carbide pretreatment before DC are summarized in Table 2.

**Table 2.** Summary of the works on tungsten carbides pretreatment before DC.

| Reference | Pretreatment type | Process |
|---|---|---|
| [21–46] | | Murakami reagent ($K_3[Fe(CN)_6]$:$KOH$:$H_2O$ in a ratio of 1:1:10) and then in a solution of peroxide mono sulfuric acid $H_2SO_5$ (Caro's acid) |
| [37,38] | Selective dissolution of cobalt with aggressive chemicals | Combined heat treatment (heating WC-Co at 1000 °C in an atmosphere of 0.25% $CH_4/H_2$) with subsequent selective removal of cobalt in Caro's acid |
| [25,39] | | Treating with microwave $CO_2$–$O_2$ plasma at 900 °C. As a result of plasma action, $CoWO_4$ and $WO_3$ compounds formed and removed in an alkaline solution |
| [42] | | etching for 1 h in a modified mixture (65 vol.% $HNO_3$ + 32 vol.% $H_2O_2$) in an ultrasonic bath |
| [43] | | Ultrasonic treatment in a mixture of 10 mL 98% $H_2SO_4$ + 30 mL 38% $H_2O_2$ |
| [25,37,38,52–55,58–60] | | The laser and plasma irradiation |
| [49] | | The impact hydrotreatment |
| [50,51] | | The shot blasting with solid particles ($SiO_2$ or $Al_2O_3$) and ultrasonic cavitation |
| [61] | Physical-mechanical | The sandblasting + acid etching |
| [50] | | The mechanical treatment with diamond powders |
| [56] | | The chemical etching, sandblasting, and creation of a carbide-forming metal layer |
| [57] | | The chemical etching pretreatment, laser etching, and acid treatment pooled plasma and chemical etching of WC-Co, followed by laser etching |
| [27,52,80,85–87] | | The Cr/CrN/Cr buffer layer, Cr, and CrC |
| [71] | | The buffer layers TiN, TiC, (TiSi)N, $Si_3N_4$, a-SiC, and a-$SiC_xN_y$ |
| [21] | | The CrN, ZrN, NbN, and TaN with a thickness of 1 μm |
| [16–20,67,77] | | W |
| [78,79] | | Al |
| [82] | | $Si_3N_4$ |
| [81] | | NbC |
| [21,81,83] | | TiN, Ti(CN) |
| [21,106] | The buffer layers formation | Ta |
| [108] | | The mixtures of $Ta_2C$ and TaC |
| [110] | | TaN/ZrN |
| [89] | | The magnetron sputtering CrN and TiN layers |
| [48] | | The buffer layers of $TiB_2$ with a thickness of about 0.1–1 μm |
| [97,98,102,103] | | SiC buffer layer |
| [96] | | HfC/HfC-SiC |
| [95] | | The gradient coatings from Mo/$Mo_2C$ |
| [75,104] | | The amorphous buffer films, a-$SiO_2$ and a-SiC |

### 9. CVD Diamond Coatings Deposition by the Hot Thread Method

In most publications devoted to the application of DC to WC-Co tools, the coating of carbide tools is carried out by the hot thread method. First of all, this is due to economic considerations since hot-thread installations allow the diamond coating to be applied to a sufficiently large area and reduces the cost of products [6,10,77,90,104,111–120]. However, the inherent disadvantages of the hot thread method do not allow to use of it cutting tools with acceptable quality for several unique applications.

So, the high-temperature alloy thread is heated to a temperature of 2000 °C during deposition and is a source of DC pollution. A decrease in the filament temperature inevitably leads to a reduction in the coating's deposition rate and, as a rule, to an increase in the proportion of non-diamond carbon in the DC. In addition, the service life of the thread is short, which reduces the productivity of the growth installation.

It should also be noted that the hot-thread installations have significant limitations on the possibility of multi-layer and gradient DC synthesis operational control [10]. The use of double-layer DC significantly increases the performance characteristics of the cutting tool. The nanocrystalline top layer provides a slight roughness, low coefficient of friction values, and high bending strength. The lower layer is usually deposited in the microcrystalline diamond growth mode, and such layers have better adhesion to WC-Co and high hardness and elasticity. In addition, two-layer DC, compared to single-layer, have a higher thermal conductivity and increased resistance to cracking. Currently, two-layer DC is deposited on WC-Co by the hot thread method. At the same time, the grain size can be changed by adjusting the bias voltage or by simultaneously changing the methane concentration and the gas mixture pressure. However, judging by the results [10], when the deposition parameters change, the formation of a layer of $sp_2$-carbon usually occurs on the boundary between micro- and nanocrystalline DC, which can negate all the advantages of two-layer DC deposited by the hot thread method.

### 10. Diamond Coating by CVD with Activation of the Gas Phase by Plasma in a Microwave Discharge

Recently, there has been a clear trend in the broader application of DC deposition technology to carbide tools in microwave plasma. The hot thread method is noticeably inferior to this technology in terms of parameters, such as gas consumption, growth rate, high purity of the gas mixture, and the possibility of alloying DC in the deposition process. However, a serious problem in ensuring same thickness and grain size of the diamond film during the deposition by the microwave plasma method remains difficult. The successful deposition of DC with a thickness of 20 to 40 μm from mixtures of methane (0.5–1.0 vol.%) and hydrogen was performed for the first time in refs [121,122]. The substrate temperature was 700–800 °C, the pressure in the chamber was 50–70 Torr, and the deposition rate was 1–2 μm/h. Work on the deposition of DC on an instrument from WC-Co was also carried out in the USA [35]. In ref [84], the deposit of DC on WC-Co (3 and 6% Co) was also carried out in microwave plasma.

In contrast, the deposition temperature was 900 °C, the proportion of methane in hydrogen was 0.5%, the pressure was 35 Torr, and the deposition time was 15–30 h. In the work cycle [53,56,123], the deposition of DC was carried out at a substrate temperature of 700–750 °C and pressure in the chamber of 80 Torr. The proportion of methane in the gas mixture was 1%, the deposition time was 6 h, and the deposition rate of DC was 1.5 μm/h. In refs [124–126], when nanocrystalline DC was deposited on WC-Co in microwave plasma, nitrogen was added to the gas mixture, and smooth and solid (80 GPa) DC with a thickness of ~10 μm was obtained. These coatings were not inferior to the DC obtained by the hot thread method. However, they did not exceed their operating parameters.

The advantages of the DC deposition method on a WC-Co instrument in microwave plasma were realized only in the last 2–3 years when microwave plasma began being used for the deposition of buffer layers and DC in a single process [127,128]. The variable buffer layers composition $WCoB/CoB$, $W_2CoB_2/WCoB$, $W_2CoB_2$, and others formed on the WC-

Co substrate in microwave plasma when 0.6 vol.% diborane $B_2H_6$ was added to hydrogen. During DC deposition, the substrate temperature varied from 600 to 1000 °C, the pressure in the chamber was 50 Torr, the proportion of methane and nitrogen in hydrogen was 9 and 1%, respectively, and the deposition time was from 0.5 to 4 h. The hardness and modulus of elasticity of DC were only 15 and 400 GPa, respectively, at low deposition temperatures (600 °C). This hardness is associated with a large proportion of non-diamond carbon. An increase in the substrate temperature during deposition improved the parameters of the DC. The hardness and modulus of the coating's elasticity increased to 32 and 600 GPa at a deposition temperature of 800 °C and at 1000 °C, up to 600 and 700 GPa, respectively. Due to the addition of nitrogen to the gas mixture and a sufficiently high content of methane within it (9%), a smooth nanocrystalline DC with a noticeable proportion of non-diamond carbon was deposited in the microwave plasma. In ref [95], WC-Co substrates with buffer layers of molybdenum dicarbide $Mo_2C$ were used to precipitate DC from microwave plasma, and in ref [96], diffusion layers of HfC/HfC-SiC were used.

In the works [16–20], samples of a new type of superhard tool with significantly higher wear resistance were created based on DC, intended for use in the production of parts and structures made of hard-to-process composite materials. The main problem of DC deposition in microwave plasma was solved due to the creation of inhomogeneities in the working chamber of methane concentration in the plasma volume and temperature along the perimeter of the substrate, the so-called edge effect.

In refs [16,17], an original design of the substrate holder was proposed, which makes it possible to implement a deposition mode of homogeneous DC properties and eliminates direct heating of high-aspect (height-to-diameter ratio) substrates in a microwave plasma reactor. The transition to indirect heating made it possible to reduce the edge effect and provided protection for the edges of the substrates from overheating. The coatings with uniform morphology were obtained both in the center and in the periphery. In refs [17,18], it established that the temperature affecting the diamond growth rate and the grain size of polycrystalline DC significantly depends on the distance between the plasma-forming surface of the substrate holder, the growth surface of the substrate, and the temperature range for the microwave reactor "ARDIS-100" with a power of 2.9 kW (determined with an accuracy of 20 °C). This technology enables the application of DC to cutters made of WC-6% Co, with an aspect ratio of 0.45. As a result, the roughness in the center and periphery of the coating differed by no more than 10% (Figure 7) [17]. A practical method of regulating the temperature of the growth surface and controlling the structure of polycrystalline DC is an operational change in the distance Δh between the substrate and the substrate holder during synthesis in microwave plasma, with the position of the cooled substrate holder unchanged.

In refs [19,20], the processes stimulating the effect of the secondary nucleation and reducing the size of diamond crystallites were used when nitrogen was added to the gas mixture during DC deposition in the plasma chemical reactor ARDIS-100 (2.45 GHz, 5 kW) with a variable supply of the gas mixture $CH_4/H_2$ and $CH_4/H_2/N_2$ at different stages of growth. In the ARDIS-100, the gas mixture is supplied from above. Typical gas flow values are 0.2–1 L/min. The DC deposition process was carried out at a total gas consumption of 500 std $cm^3$/min and a chamber pressure of 65 Torr at a microwave power of 3.0 kW. The substrates in the synthesis process were at a temperature of 750 °C, and the deposition rate was 2–3 μm/h. The methane concentration at the time of plasma ignition was 0% and gradually increased to 4% in ~2 min, with a simultaneous increase in the microwave radiation power and substrate temperature (Figure 8).

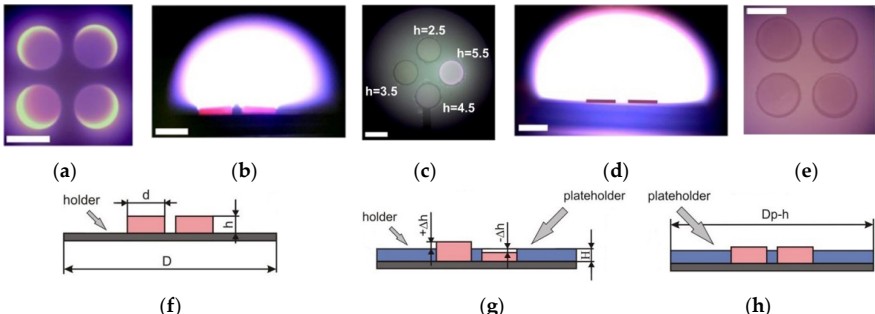

**Figure 7.** Top view (**a**) and side view (**b**) of the samples in an open substrate holder during the first series of growth of CVD diamond; top view (**c**) of the substrates with different heights in the holder holes during the second series of growth of CVD diamond; and side view (**d**) and top view (**e**) of substrates with equal heights in the holder during the third series of growth of CVD diamond. Schematic views of the arrangement of the substrates in the open holder (h/d = 0.45) during the first series of growth of CVD diamond (**f**); the same for the close-type holder with substrates of different heights during the second series of growth of CVD diamond (**g**), and the same for the open holder (h/d = 0.45) during the third series of growth of CVD diamond (**h**). The plate holder diameter is Dp–h = 57 mm, and the height is H = 3.5 mm. White scale bars size in the lower left corner of sub-figures (**a**–**e**) is 10 mm [17].

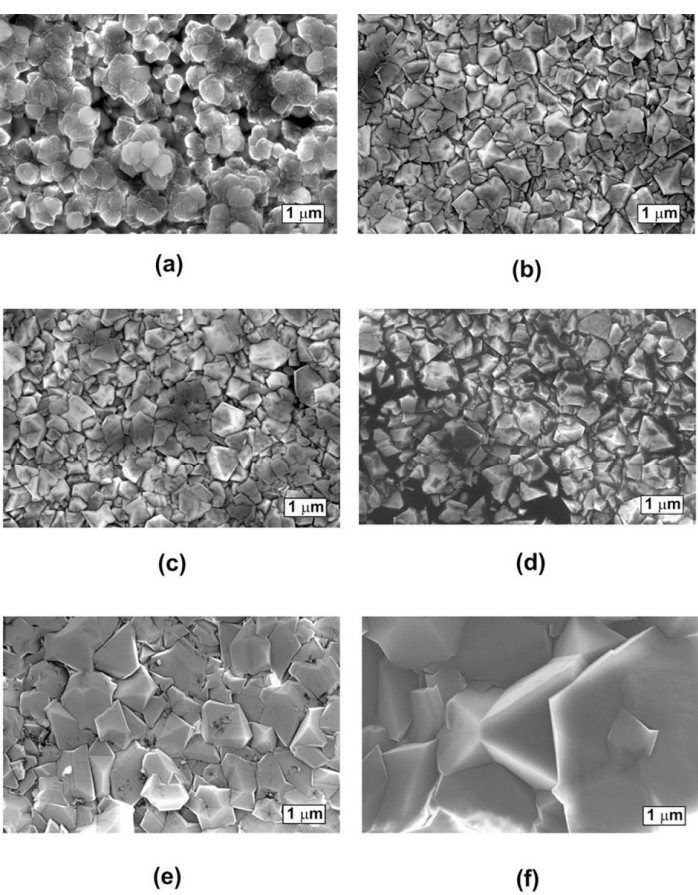

**Figure 8.** SEM images of diamond films deposited at different temperatures and distances between the substrate and plate holder surfaces: (**a**) Δh = −1 mm, T = 700 °C, half of the radius; (**b**) Δh = 0 mm, T = 740 °C, center; (**c**) Δh = 0 mm, T = 740 °C, half of the radius; (**d**) Δh = 0 mm, T = 740 °C, periphery; (**e**) Δh = 1 mm, T = 760 °C, half of the radius; and (**f**) Δh = 2 mm, T = 870 °C, half of the radius. Micrographs (**a**,**e**,**f**) relate to the second CVD series, and micrographs (**b**–**d**) relate to the third CVD series [17].

The temperature range ensuring the uniform deposition of DC on WC-6% Co substrates in the ARDIS-100 reactor is 740–760 °C at a power of 2.9 kW and a distance of $0 \leq \Delta h < 1$ mm. The change in the DC $R_a$ roughness from the center to the periphery does not exceed 15%.

This solution allowed it to preserve the nuclei of diamond particles from burning out and significantly increased their density to ~$10^9$ cm$^{-2}$ or more. The multi-layer coatings were successive layers of micro- and nanocrystalline diamonds. The synthesis of microcrystalline layers was carried out without the addition of nitrogen. The concentration of $N_2$ in the total gas flow was maintained at the level of 4% during the deposition of nanocrystalline layers. According to RAMAN spectroscopy data [20], an intermediate layer with a high concentration of the sp2 phase does not form at the boundary of the micro- and nanocrystalline diamond layers as often as it happens with the growth of gradient layers by the hot thread method, which positively affects the performance characteristics of the tool.

In ref [129], DC deposition in microwave plasma was carried out on buffer layers of a diamond/β-SiC composite grown in the same plant, and the proportion of β-SiC varied from 12 to 68%. The DC layers deposited on the diamond/β-SiC composite demonstrated good adhesion and low-friction coefficients.

## 11. The Diamond-Coated Carbide Tools for Processing Polymer Composite Materials

In the drilling processes of polymer materials reinforced with carbon fibers (CFRP), conventional cutting tools wear out quickly due to the abrasive effect of carbon fibers on the cutting surfaces of the instrument. Figure 9 shows the wear pattern of the cutting edge, demonstrating the formation of a spout with a larger width than the original tool [84]. The load on the cutting edge is very high in the absence of coolant; therefore, the requirements for the DC are very stringent. The primary defects of the holes processed with a diamond-coated tool are the ejection and destruction of the fiber, cracking of the matrix, and stratification of layers and burrs at the outlet [130–135].

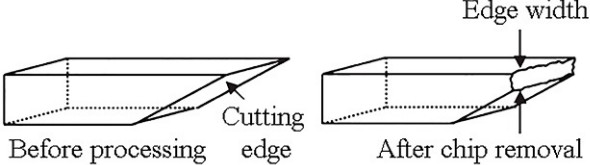

**Figure 9.** Diagram of the wear of the cutting edge showing the formation of a nose with a wider width than the original tool [84].

When cutting CFRP, a positive effect of DC was noted due to their high hardness and low coefficient of friction [136]. In refs [137,138], the results of tests for cutting the edges of carbon fiber plastics are presented, from which it follows that a diamond-coated tool has better wear resistance than a traditional cutter. However, CFRP delamination occurs when the feed rate of the diamond-coated instrument increases. In ref [4], tool wear was studied when drilling CFRP with various drills, including a DC drill, and it was found that a diamond-coated drill cutting length is 80% longer than a conventional tool.

Boron or silicon-doped DC deposited by the hot thread method on chemically treated WC-Co drills are optimally suited for processing CFRP products [139]. Rockwell hardness measurements have shown that doping with boron and silicon on the surface of a hard alloy can increase the adhesion between DC and WC-Co. The Si-doped drills have better adhesive strength and higher performance and wear resistance than drills without DC when processing CFRP materials with a coated tool. The positive effect of diamond doping with silicon is the reduction of nanoscale diamond crystallites of the coating and the formation of a smoother surface with a low coefficient of friction during cutting [139].

In ref [140], the feeding force and delamination processes were investigated during the operation of drilling carbon plastics with a special geometry tool. It was found that it is necessary to reduce the feed rate to reduce delamination. In ref [141], a mechanical model

proposed that it is possible to predict the cutting force when drilling CFRP composite laminates. In ref [142], a physical model was proposed that predicts the critical feed force at the beginning of stratification when drilling CFRP and allows for analyzing the stratification process of drills with different geometries.

In ref [131], the dependence of the stratification processes of the processed material on the cutting speed and feed rate was studied. It shows that an increase in the cutting speed and feed increases the stratification. A comparative analysis of the cutting characteristics for step drills when drilling small diameter holes in CFRP showed that the main factors affecting the service life of a diamond tool are its geometry and feed rate [143]. In ref [144], it was found that when drilling holes in CFRP, the main wear factors are grinding, abrasion, and adhesion of the DC.

Diamond-coated tools with high service properties are the primary tools for processing CFRP [145]. They provide a longer service life and better surface quality [146]. Tools with CVD DC combine record-breaking DC properties with high geometric flexibility. High hardness and wear resistance, low coefficient of friction, and good thermal conductivity reduce cutting forces and heating, which increases the tool's life [147]. However, insufficient adhesion of the DC can reduce its service life. In ref [148], optimal cutting characteristics of polyether ether ketone are composites with a 30% volume fraction of carbon fibers with a CVD.

Figure 10 shows data on the continuity of the tool with and without DC when turning a metal matrix composite A-390 [17]. As can be seen, the DC noticeably increases the resource of the instrument. Figure 11 shows the appearance of the 10th exit hole, treated with uncoated and CVD DC with a grain size of ~3 μm, obtained by the hot thread method [149]. Due to the excellent wear resistance, the tool's durability with DC is much higher.

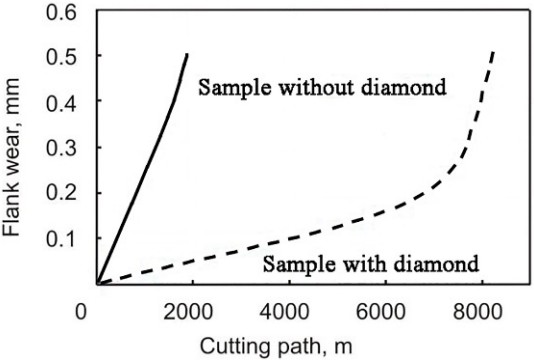

**Figure 10.** Period of service of the tool with and without diamond coating under the turning of the A-390 type metal-matrix composite with 18% SiC [17].

According to the form of wear of the drill with DC when drilling CFRP, it was established that the observed wear was mainly caused by the abrasive behavior of the reinforcing fiber. The pressure force on the WC-Co drill with DC during the drilling process is less than on the uncoated instrument due to the low coefficient of friction. It shows that the best quality of the holes, especially the inner surface of the hole, is obtained using drills with DC, and it concluded that diamond CVD coatings could solve the problems that arise when processing CFRP [149].

Comparative studies [150] of orthogonal cutting processes with a WC-6% Co tool with and without a CVD nanocrystalline DC with a thickness of 5–6 μm and an average grain size of 50–100 nm showed that at low processing speeds, the cutting force for a tool with and without DC differed slightly, which is due to a large radius of the cutting edge for a tool with a DC, weakening the advantages of a smooth DC. The cutting force decreased compared to a conventional tool with increased cutting speed due to low friction and good wear resistance.

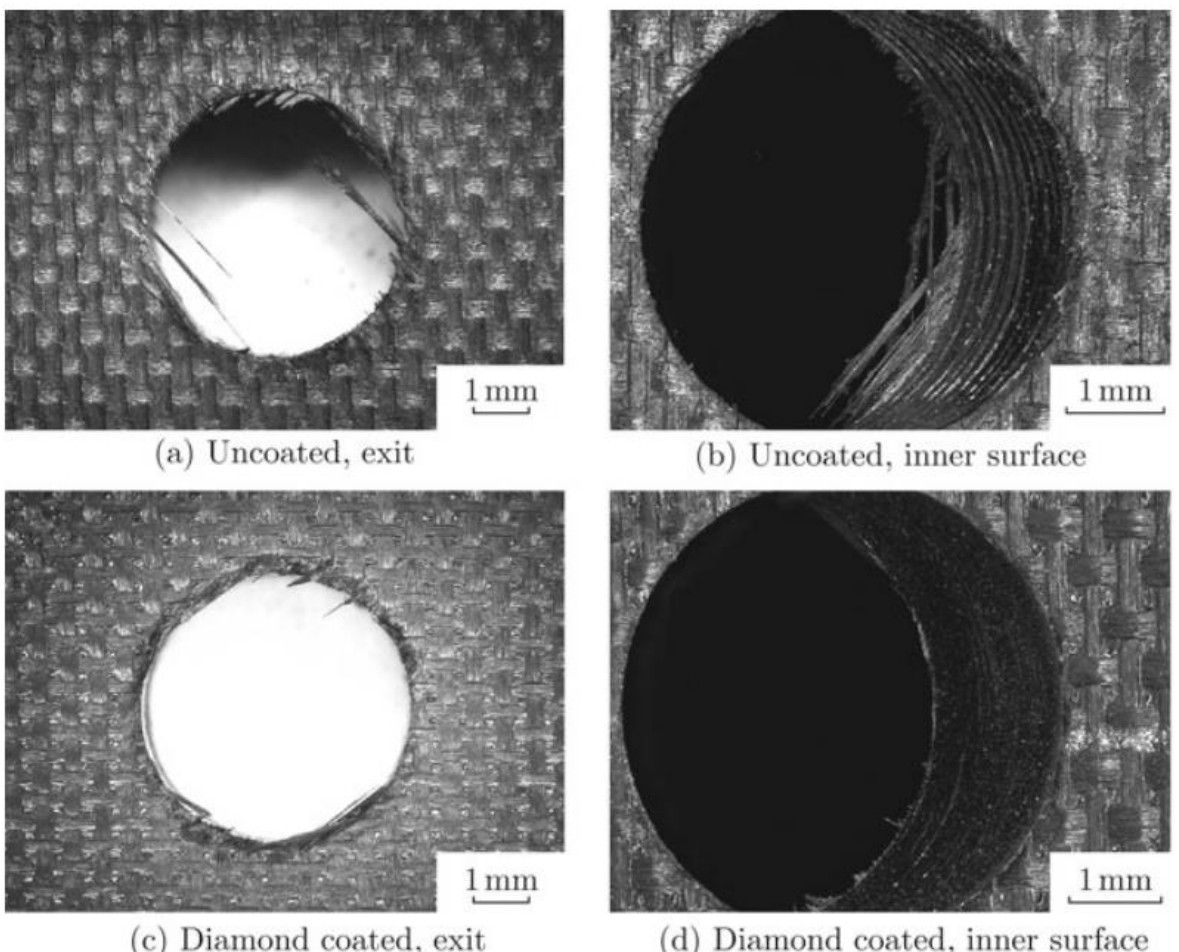

**Figure 11.** Images for the tenth hole machined by uncoated and diamond-coated drills: (**a**) The outcome made by an uncoated drill, (**b**) the inner surface of the hole made with a drill without coating, (**c**) the outcome made by a diamond-coated drill, and (**d**) the inner surface of the hole made with a drill with diamond coating [149].

## 12. Conclusions

Despite many experiments on research and development of technology for creating a durable tool with diamond coating, the data obtained should be compared with a great deal of caution since different authors used equipment of varying quality and had other technological qualifications. Therefore, in several cases, the same technological solution is evaluated differently in other works. The literature data on the deposition of diamond coatings on a cutting tool shows that the direct deposition of high-quality DC on WC-Co is a highly complex process. Co catalyzes the formation of a brittle graphite sublayer between diamond and substrate and the mismatch of the CTE of diamond and WC-Co.

Currently, various methods of pretreatment of the substrate surface are used, including chemical etching, mechanical hardening, plasma treatment, laser ablation, application of buffer layers between the substrate and the coating, and various ways to increase the nucleation density of diamond particles.

Tribological characteristics of the tool with DC depend on the microstructure and roughness of the coating. The adhesion, hardness, and wear resistance of the DC increase after several stages of WC-Co pretreatment before deposition of the DC, leading to the creation of a multi-layer, three-dimensional interface.

Murakami reagent and Caro acid are widely used for the chemical etching of WC-Co. The choice of the optimal chemical etching mode for each type of substrate with different concentrations of cobalt, the buffer layer, and DC deposition technology, as well as the

operating conditions of the diamond tool, is individually selected. The same applies to the choice of chemical composition, method, mode of deposition, and geometric parameters of buffer layers. The adhesion of the DC is determined by the structure and adhesive characteristics of the buffer layers. So the choice of optimal materials and technology for buffer layers obtaining depends on the type and mode of operation.

The widespread technology of DC deposition by the hot thread method, which provides a low cost of manufacturing a diamond tool, is inferior to the process of plasma deposition using a microwave discharge. The friction coefficient of a diamond-coated tool can be reduced to 0.07 compared with 0.6 for an uncoated tool by reducing the consumption of plasma-forming gas, increasing the growth rate of the coating, as well as significantly expanding the possibilities of nitrogen doping from the gas phase during deposition and formation of two- and multi-layer gradient coatings. The microwave plasma makes it abandon consumable electrodes, increases the purity in the discharge chamber, and contributes to obtaining a perfect structure of diamond coatings. This is especially relevant when creating tools for high-precision mechanical processing of polymer composite materials reinforced with fibers with high elastic properties, which imposes strict requirements on the strength properties of the diamond coating.

**Author Contributions:** Conceptualization, E.A.; methodology, A.K.; investigation, V.R.; funding acquisition, A.B. visualization, D.S.; conceptualization, V.K.; project administration, S.G.; writing—review and editing S.F. All authors have read and agreed to the published version of the manuscript.

**Funding:** This research was carried out with the financial support of the Russian Science Foundation within the framework of scientific project No. 22-19-00694.

**Data Availability Statement:** The data presented in this study are available on request from the corresponding author.

**Conflicts of Interest:** The authors declare no conflict of interest.

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
