# Peer review of "Technology Features of Diamond Coating Deposition on a Carbide Tool"

_carbon, 2022_

Round 1

Reviewer 1 Report

The paper collects the literature concerning the different procedures proposed for obtaining a coating of WC-Co with DC with the desired mechanical properties. The subject is of great technological interest, but unfortunately the paper is very difficult to read due to the poor quality of the English language and the organization. I would suggest, in addition to the correction of the English, a much more schematic approach: for each step or type of material to be treated, a scheme with the various treatments proposed (of which the positive and problematic aspects would be presented, including the contradictory results reported in the literature) could be prepared .

Moreover, in the Introduction I suggest to indicate which are the properties (at least with numerical indications) that the finished material must have, hinting a little more extensively on applications in a less generic way.

Author Response

Dear reviewer!

Thank you so much for your interest in our review.

We have tried to take into account your requirements regarding the English language and the indication of some properties of diamond instrument in the introduction. Allow us not to change the structure of the article. Now it shows various techniques that allow to obtain sufficient adhesion of the diamond coating to the carbide substrate.

Reviewer 2 Report

This paper provided a comprehensive review on how to increase the interface bonding between diamond and WC. The product is planned to be used for machining carbon fiber reinforced composite materials.

The technical content falls into the scope of the journal.

Some minor corrections are suggested.

On line 253, "In [27], it shown that " should be corrected as "In [27], it has been shown that ".

The reference format has to be checked. The volume sign "v" is not necessary for MDPI journals; only the numbers are needed.

On line 308, check " a-C" and make sure it is correct.

On line 327, check the unit "mkm" and make sure it is correct.

Author Response

Dear reviewer!

Thank you so much for your interest in our review.

We have tied to take into account your requirements regarding the English language and some correction in the text.

Reviewer 3 Report

Dear Editor:

Thank you for giving me this chance to review this paper, and the evaluation of manuscript entitled “Technology features of diamond coating deposition on a carbide tool” is shown as follows.

1.The abstract and introduction are important for the attraction of the published article. But the authors failed to write abstract and introduction clearly and concisely, especially introduction. Therefore, try to get your ideas into shape and delete some unnecessary descriptions, at a glance, to capture the attention of a wide readership.

2.Note the units and some misrepresentations.

For example: Paragraphs 85,130,158,237,280,346,361,529,697, etc.

3.The chapter of the article needs to be revised. Some of it overlaps.

For example, the contents of section 678 should be included under 5. It is suggested that the order of description should be “Buffer layers based on silicon compounds, Buffer layers based on tungsten and tantalum and Two-layer and gradient buffer layers”.

4.The description in paragraphs 260-270 should be chemical modification rather than Physical-mechanical methods.

5.Currently, various methods of pretreatment of the substrate surface are used, including chemical etching, mechanical hardening, plasma treatment, laser ablation, application of buffer layers between the substrate and the coating, and various ways to increase the nucleation density of diamond particles. These methods have been described in previous research reviews. Is it possible for the author of this paper to propose a treatment method with cutting-edge research to provide a reference for the future research direction?

6. Several typos and grammatical errors in the manuscript. Authors have to check and correct the manuscript thoroughly.

Author Response

see file

Round 2

Reviewer 1 Report

The authors improved the paper by taking into account the reviewers' suggestions. 

The paper remains not easy to read due to the large number of processes listed in. I'd suggest to the authors to add some tables to summarize the different preatments, and different technologies for deposition inserting advantages and disadvantages of the various approaches.

In this way the interested reader, but not an expert in the specific topic,  can enter and benefit from reading the paper.

page 2, line 48: mkm, what does it mean? line 58: 600-900°C

page 4, line 137 H2O2, please correct the formula

Author Response

Dear reviewer!

Thank you for your work.

On your advice, I added a table where I summarized the main methods of pretreatment.

S. Fedorov

Reviewer 3 Report

Accept in present form

Author Response

Thank you so much for your work